# Nuclear Beclin 1 Destabilizes Retinoblastoma Protein to Promote Cell Cycle Progression and Colorectal Cancer Growth

**DOI:** 10.3390/cancers14194735

**Published:** 2022-09-28

**Authors:** Yang Pan, Zhiqiang Zhao, Juan Li, Jinsong Li, Yue Luo, Weiyuxin Li, Wanbang You, Yujun Zhang, Zhonghan Li, Jian Yang, Zhi-Xiong Jim Xiao, Yang Wang

**Affiliations:** 1Center of Growth, Metabolism and Aging, Key Laboratory of Bio-Resource and Eco-Environment, Ministry of Education, College of Life Sciences, Sichuan University, Chengdu 610065, China; 2State Key Laboratory of Biotherapy, Sichuan University, Chengdu 610041, China

**Keywords:** Beclin 1, retinoblastoma protein, MDMX, cell cycle, colorectal cancer

## Abstract

**Simple Summary:**

The role of autophagy core-protein Beclin 1 in colorectal cancer (CRC) development remains controversial. Here, we show that nuclear Beclin 1 is up-regulated in CRC with a negative correlation to RB protein expression. Silencing of *BECN1* up-regulates RB expression resulting in cell cycle G1 arrest and inhibition of xenograft tumor growth independent of p53. Ablation of *BECN1* facilitates MDM2–MDMX complex formation to promote MDMX polyubiquitination and degradation, consequently leading to RB protein stabilization. These results reveal that nuclear Beclin 1 can promote CRC growth through modulation of RB protein stability and imply that nuclear Beclin 1 may be a prognostic indicator in human colorectal cancer.

**Abstract:**

Autophagy is elevated in colorectal cancer (CRC) and is generally associated with poor prognosis. However, the role of autophagy core-protein Beclin 1 remains controversial in CRC development. Here, we show that the expression of nuclear Beclin 1 protein is upregulated in CRC with a negative correlation to retinoblastoma (RB) protein expression. Silencing of *BECN1* upregulates RB resulting in cell cycle G1 arrest and growth inhibition of CRC cells independent of p53. Furthermore, ablation of *BECN1* inhibits xenograft tumor growth through elevated RB expression and reduced autophagy, while simultaneous silencing of *RB1* restores tumor growth but has little effect on autophagy. Mechanistically, knockdown of *BECN1* promotes the complex formation of MDM2 and MDMX, resulting in MDM2-dependent MDMX instability and RB stabilization. Our results demonstrate that nuclear Beclin 1 can promote cell cycle progression through modulation of the MDM2/X-RB pathway and suggest that Beclin 1 promotes CRC development by facilitating both cell cycle progression and autophagy.

## 1. Introduction

Colorectal cancer (CRC) is the third most commonly diagnosed malignancy and the second leading cause of cancer death. The CRC is attributable to environmental risk factors including obesity, alcohol drinking, smoking, and inflammatory bowel disease. Inactivation of the tumor suppressor genes such as *APC*, *TP53*, and *RB1* (the retinoblastoma gene) are critically important for CRC development [1]. The *RB1* inactivation yields a mouse model of malignant colorectal cancer [2]. *RB1* encodes prototypical tumor suppressor retinoblastoma protein (RB), which plays pivotal roles in the regulation of the cell cycle; DNA replication; DNA damage response; and repair, senescence, apoptosis, and differentiation [3]. RB binds to the E2F family of transcription factors and forms an active transcription repressor complex that blocks the expression of genes involved in DNA replication and cell cycle progression. The cell-cycle-dependent kinases (CDKs)-mediated hyperphosphorylation of RB proteins leads to disruption of RB–E2F complexes and consequently to cell cycle progression [3]. Previous studies from us and others have shown that MDM2 (the double mouse minute 2 homolog) directly binds to the C-pocket of RB, leading to disruption of RB–E2F interaction and acceleration of RB protein degradation [4,5]. MDMX (also called MDM4), a homolog of MDM2, can also bind to and promote RB degradation in an MDM2-dependent manner, thereby promoting tumor growth [6].

Autophagy is an evolutionarily conserved mechanism that is critical in maintaining cellular homeostasis by degrading unnecessary or dysfunctional organelles and proteins. Autophagy can either promote or inhibit tumor growth in a context-dependent manner [7]. Autophagy appears to be specifically upregulated in colorectal cancer and is generally associated with poor prognosis and drug resistance [8], but autophagy studies in CRC show conflicting results, particularly concerning chemotherapy resistance of colorectal cancer [7,9]. The autophagy-related gene *BECN1* (i.e., *ATG6*) is critically important in the initiation and maturation of autophagosomes. *Becn1*${}^{+/-}$ mice show an increased incidence of spontaneous tumors, including B cell lymphomas, lymphoblast cell lymphoma, hepatocellular carcinomas, and lung adenocarcinoma, while *Becn1*${}^{-/-}$ cause the embryonic lethality [10]. The role of *BECN1* in colorectal cancer development remains controversial. Knockdown of *BECN1* has been reported to suppress epithelial-mesenchymal transition and to attenuate invasiveness of CRC cells [11], while ectopic expression Beclin 1 leads to growth retardation of the CRC cells [12]. Whether *BECN1* can regulate the cell cycle progression of colorectal cancer cells remains unknown.

Beclin 1 is localized in both cytosol and nuclei. Beclin 1 contains a leucine-rich nuclear exporting signal motif leading to a CRM1-dependent nucleocytoplasmic transport, which is required for autophagy [13]. Cytoplasmic Beclin 1 functions as a core protein in class III PI3K complexes that mediate different membrane trafficking events in autophagy [14], while the function of nuclear Beclin 1 is largely unknown.

In this study, we show that the expression of nuclear Beclin 1 is upregulated in colorectal cancer with a negative correlation to RB protein expression. Silencing of *BECN1* upregulated RB protein expression resulting in cell cycle G1 arrest and growth inhibition of colorectal cancer cells. Furthermore, ablation of *BECN1* inhibited xenograft tumor growth, which was reversed by simultaneous silencing of RB. Mechanistically, we show that the knockdown of *BECN1* promotes MDM2 and MDMX interaction, resulting in MDMX protein instability and RB protein stabilization.

## 2. Materials and Methods

### 2.1. Cell Culture and Reagents

HCT116, HCT116 p53${}^{-/-}$, A549, and HEK293FT cells were cultured in DMEM medium (GIBCO, Rockville, MD, USA) supplemented with 10% fetal bovine serum (Hyclone, Logan, UT, USA) and penicillin (100 U/mL)/streptomycin (100 μg/mL) (Hyclone). Caco-2 cell was cultured in DMEM medium (GIBCO, Rockville, MD, USA) supplemented with 20% fetal bovine serum (Hyclone, Logan, UT, USA) and penicillin (100 U/mL)/streptomycin (100 μg/mL) (Hyclone). Cells were maintained at 37 ${}^{\circ }$C in a humidified 5% CO${}_{2}$ incubator. MG132 (S2619), cycloheximide (S7418), and puromycin (S7417) were purchased from Sigma-Aldrich (St. Louis, MI, USA). Diphenylterazine (HY-111382) was purchased from MCE (Shanghai, China).

### 2.2. Plasmids, Lentiviral Infection, and RNA Interference

Recombinant lentiviruses used in this study, including pcDNA3.1-Flag-Beclin 1, pcDNA3.1-Flag-Beclin 1-ΔBH3, pcDNA3.1-Flag-Beclin 1-ΔBH3ΔCCD, pcDNA3.1-Flag-Beclin 1-ΔECDΔC, pLVX-Beclin 1, pLVX-Beclin 1${}^{L184A+L187A}$, pLVX-Antares2 [15], pLVX-MDMX, and pLVX-MDMX${}^{C437A}$, were constructed and verified by direct DNA sequencing, Lentiviral-based short hairpin RNAs (shRNAs) specific for *BECN1*, *RB1*, and *GFP* (control) were cloned into pLKO.1-TRC as previously described [16]. Targeted sequences for shRNA are listed in Appendix A.

HCT116 p53${}^{-/-}$ cells were transfected with siMDM2, siMDMX, or a scrambled siRNA using Lipofectamine${}^{\mathrm{TM}}$ 3000 (Invitrogen). Cells were collected 24 h after transfection and cell lysates were subjected to western blot analyses as previously described [4]. Targeted sequences were listed in Appendix A.

### 2.3. Western Blot, Immunofluorescence (IF), and Immunohistochemistry (IHC) Analyses

Western blot, IHC, and IF analyses were performed as previously described [16]. For western blot analyses, antibodies specific for Beclin 1 (3495), LC3 (2775), Histone 3 (9715S), HA (2367), and Flag (14793) were purchased from Cell Signaling Technology (Danvers, MA, USA); antibodies for p21 (CY5543), GAPDH (AB0037) and MDM2 (CY3621) were purchased from Abways (Shanghai, China); antibody specific for MDM2 (sc-965) was purchased from Santa Cruz Biotechnology (Dallas, TX, USA); antibody specific for His (230001) was purchased from Zen BioScience (Chengdu, China); antibody specific for MDMX (A300-287A) was purchased from Bethyl Laboratories (Waltham, MA, USA); antibody specific for RB (554136) was purchased from BD Biosciences (San Jose, CA, USA).

For IHC analyses, antibodies for Beclin 1 (ab62557), p62 (ab56416), LC3B (ab48394), and Ki67 (ab15580) were purchased from Abcam (Cambridge, MA, USA); the antibody specific for RB was (554136) purchased from BD Biosciences (San Jose, CA, USA); peroxidase (HRP)-conjugated secondary antibody (DS-0004) was purchased from Zhongshan Golgen Bridge Biological Technology (Beijing, China). For quantitative analysis, tissue slides were scanned through NanoZoomer (Hamamatsu, Japan), and the scanned images were subjected to analyzing average optical density (AOD) [16] using QuPath [17]. Human colorectal cancer tissue microarrays (TMA) (HColA030PG06) were purchased from Outdo Biotech (Shanghai, China).

For IF analyses, an antibody specific for Beclin 1 (ab62557) was purchased from Abcam (Cambridge, MA, USA) and an antibody specific for RB (544136) was purchased from BD Biosciences (San Jose, CA, USA). Rhodamine (TRITC)-conjugated AffiniPure Donkey Anti-Rabbit IgG (711-025-152) and Fluorescein (FITC)-conjugated AffiniPure Donkey Anti-Mouse IgG (715-095-150) were purchased from Jackson Immuno Research (West Grove, PA, USA).

### 2.4. Flow Cytometry Analysis

Cells were trypsinized, washed with cold PBS, and fixed in 70% ethanol at 4 ${}^{\circ }$C overnight. Cells were stained with 50 μg/mL propidium iodide supplemented with 100 μg/mL RNase A at 37 ${}^{\circ }$C in the dark for 1 h. Cells were then subjected to flow cytometry analysis by FACScan Flow Cytometer [6] (Becton Dickson, San Jose, CA, USA), and the data were analyzed using Cell Quest software (Becton Dickson).

### 2.5. Quantitative PCR (qPCR) Analysis

Total RNA was extracted using NucleoSpin^®^ RNA Plus kit (740984, MACHEREY-NAGE), followed by reverse transcription using ReverTra Ace qPCR RT Master Mix (FSQ-201, TOYOBO). qPCR analyses were performed as previously described [16]. The specific primer sequences for qPCR were listed in Appendix A.

### 2.6. Colony Formation Assay

Aliquots of 500 HCT116 or HCT116 p53${}^{-/-}$ cells were seeded in 6-well plates and cultured for 20 days. The colonies were fixed for 15 min with cold methyl alcohol, stained with 0.1% crystal violet for 15 min at room temperature, and photographed. Colonies of >50 cells were used for quantitative counts.

### 2.7. Real-Time Cell Analysis (RTCA)

For cell index (CI) analysis, experiments were performed using an xCELLigence Real-Time Cell Analysis (RTCA) [18] DP instrument (Roche Diagnostics GmbH, Mannheim, Germany) at 37 ${}^{\circ }$C with 5% CO${}_{2}$. To measure the cell index of HCT116 or HCT116 p53${}^{-/-}$ cells in real-time, cells were seeded on gold microelectrodes embedded at the bottom of 16-well microplates (E-plates; Roche Diagnostics, Basel, Switzerland) at a density of 8.0 × 10${}^{3}$ cells/well for HCT116 or HCT116 p53${}^{-/-}$ cells. The impedance was recorded at 15 min intervals. Cell index values were evaluated by the RTCA-DP software (Roche Diagnostics GmbH, Metroplaza, Germany).

### 2.8. Mouse Xenograft Studies

All the mouse strains were kept in standard, infection-free housing conditions, with 12 h light:12 h dark cycles and 3–5 mice per cage. Animals were housed in a pathogen-free barrier environment throughout the study. All animal experiments in this study were approved by the Institutional Animal Care and Use Committee of Sichuan University (IACUC), and the operating procedures were carried out in accordance with the guidelines formulated by China’s Council on Animal Care.

For in vivo tumor progression, Antares2 [15]-expressing HCT116 p53${}^{-/-}$ (5.0 × 10${}^{6}$) or their derivatives were subcutaneously transplanted into the right front flanks of 6-week-old female nude mice (*n* = 5/group) (Model Animal Research Center of Nanjing University, China). Tumor size was measured with a caliper every other day, and tumor volume was calculated by width${}^{2}$× length × $\frac{1}{2}$. A total 0.1 mM Diphenylterazine (DTZ, MCE) (the substrates of Antares2) in 100 μL PBS was injected into tumor regions in anesthetized mice using tribromoethanol (250 mg/kg intraperitoneally (i.p.)). Mice were recovered on heat pads for 5 min and bioluminescence was subsequently imaged using a Caliper IVIS Lumina (Perkin Elmer, Waltham, MA, USA). The images were processed using the Living Image 4.5.5 software.

### 2.9. Correlation and Survival Analyses Based on Data from Clinical Specimens

The mRNA expression data and reverse phase protein array (RPPA) data of colon adenocarcinoma from TCGA COADREAD were retrieved using RTCGA package (https://rtcga.github.io/RTCGA (accessed on 10 September 2020 )). Survival analysis was conducted by survminer package (https://github.com/kassambara/survminer (accessed on 10 September 2020)) using optimal grouping determined by surv_cutpoint function. The correlation of *RB1* and *BECN1* at various mRNA levels was calculated by cBioPortal (https://www.cbioportal.org/ (accessed on 10 September 2020)) [19,20] and GEPIA (http://gepia.cancer-pku.cn/ (accessed on 10 September 2020)) [21], while their correlation at various protein levels was calculated by LinkedOmics (http://www.linkedomics.org/admin.php (accessed on 10 September 2020)) [22].

### 2.10. Statistical Analyses

Data from at least three independent experiments were presented as mean ± SD, and data from animal experiments were presented as mean ± SEM. Two-tailed unpaired Student’s *t*-test was used for comparing two groups of data. One-/two-way ANOVA with Tukey’s test was used to compare multiple groups of data. *p* values of less than 0.05 were considered significant.

## 3. Results

### 3.1. Elevated Expression and Nuclear Location of Beclin 1 Is Negatively Correlated with RB Expression in Malignant Human Colorectal Cancer Specimens

The tumor suppressor RB protein plays a pivotal role in the negative control of the cell cycle and tumor progression [23]. However, the clinic correlation between RB expression and human colorectal cancer remains controversial [24,25]. We therefore analyzed the current TCGA database for *RB1* expression in human colorectal cancer. As shown in Appendix A, low expression of *RB1* was associated with poor overall survival (OS) in human colorectal cancer patients. Furthermore, we found that high expression of Beclin 1 protein, but not mRNA, was associated with poor OS and relapse-free survival (RFS) in CRC patients (Figure 1a). To investigate the correlation between RB and Beclin 1, we then analyzed the correlation between Beclin 1 and RB protein expression of colorectal cancer patients in the TCGA-COADREAD dataset obtained from LinkedOmics [22]. Our analyses reveal a moderate negative correlation (r = −0.3245, Pearson’s correlation coefficient) between RB and Beclin 1 proteins (revised Figure 1b) in contrast to mRNA levels that showed a weak positive correlation (data from cBioPortal [19,20], GDC TCGA COAD [26], and GEPIA [21]) (Figure 1b and Appendix A). We further examined the correlation of RB and Beclin 1 protein expression using the human colorectal cancer tissue microarrays (TMA) consisting of colorectal cancer specimens at different clinical stages. As shown in Figure 1c–e, reduced RB protein expression and increased Beclin 1 expression were observed in human colorectal cancer specimens at stages II and III compared with the specimens at stage I. Notably, the expression of Beclin 1 protein was dramatically increased in the nucleus and was negatively correlated with RB protein expression.

### 3.2. Ablation of *BECN1* Leads to Cell Cycle G1 Arrest and Cell Growth Retardation Independent of p53

We then examined the effects of *BECN1* ablation on the growth of HCT116 or HCT116 p53${}^{-/-}$ cells. As shown in Figure 2a, as expected, knockdown of *BECN1* significantly inhibited autophagy, as evidenced by the upregulation of p62 and downregulation of LC3-II (Figure 2a). However, knockdown of *BECN1* had little effect on apoptosis, as evidenced by undetectable cleaved caspase3 (Figure 2a). Surprisingly, *BECN1* ablation led to robust upregulation of RB protein expression in both HCT116 and HCT116 p53${}^{-/-}$ cells (Figure 2a and Appendix A), concomitant with the reduction in MDMX protein levels, while the expression of MDM2 remained unchanged (Figure 2a). The expression of p53 and p21 were also dramatically upregulated in HCT116 cells but not in HCT116 p53${}^{-/-}$ cells. Similar observations were obtained in CRC Caco-2 cells and human lung adenocarcinoma A549 cells (Appendix A). Furthermore, knockdown of *BECN1* dramatically inhibited colony formation and growth in HCT116 or HCT116 p53${}^{-/-}$ cells (Figure 2b,c). Moreover, FACS analyses showed that knockdown of *BECN1* significantly induced the cell cycle G1/S arrest independent of p53 (Figure 2d,e). Restoration of wild-type Beclin 1 expression effectively reversed the knockdown of *BECN1*-induced altered expression of MDMX and RB, concomitant with restored autophagy, and significantly rescued G1/S arrest and colony formation (Figure 2f–h). Interestingly, while the expression of Beclin 1${}^{L184A+L187A}$—a mutant defective in promoting autophagy due to its exclusive nuclear localization [13]—failed to restore autophagy as expected, it could effectively reverse the knockdown of *BECN1*-induced altered expression of MDMX and RB, resulting in significantly reduced G1/S arrest and restored colony formation (Figure 2f–h). Together, these results indicate that silencing of *BECN1* leads to upregulation of RB protein expression, concomitant with the cell cycle G1 arrest and growth inhibition independent of p53, and importantly, nuclear Beclin 1 can impact cell growth through modulation of RB protein expression.

### 3.3. Ablation of *BECN1* Downregulates MDMX Expression to Induce RB-Dependent Cell Cycle Arrest and Inhibition of Cell Growth

To investigate whether RB plays a causal role in cell cycle G1/S arrest induced by knockdown of *BECN1*, a set of rescuing experiments was performed. As shown in Figure 3a–e, knockdown of *RB1* could significantly, but not completely, restore the colony formation, rate of cell growth, and cell cycle progression, all of which were inhibited by *BECN1* ablation. Furthermore, restoration of MDMX, but not MDMX${}^{C437A}$—a mutant defective in the RB degradation [6]—significantly reversed the expression of RB, which was upregulated by knockdown of *BECN1*, accompanied by markedly reduced G1/S arrest and largely restored colony formation or rate of cell growth independent of p53 (Figure 3f–i).

### 3.4. *Ablation of BECN1 Facilitates MDM2–MDMX Interaction to Promote MDMX Degradation Leading to RB Protein Stabilization*

Our aforementioned results show that *BECN1* ablation leads to the downregulation of MDMX and upregulation of RB. Notably, MDMX can rescue *BECN1* ablation-induced RB upregulation. Since MDMX has been shown to promote RB protein degradation [6], we postulated that MDMX might mediate Beclin 1-induced regulation of RB protein instability. To this end, we first investigated the effect of *BECN1* knockdown on the mRNA expression of *RB1* or *MDMX* and on respective protein stability. As shown in Appendix A, knockdown of *BECN1* had little effect on the steady-state *RB1* or *MDMX* mRNA levels. By contrast, knockdown of *BECN1* significantly extended the protein half-life of RB (Figure 4a,b). On the other hand, inhibition of proteasome by MG132 significantly rescued the shBeclin 1-induced downregulation of MDMX, suggesting that *BECN1* depletion leads to accelerated MDMX proteasome degradation (Figure 4c). Indeed, silencing of *BECN1* significantly facilitated MDMX polyubiquitylation (Figure 4d). We next investigated the molecular basis on which Beclin 1 impacts MDMX polyubiquitylation and RB protein stability. We have previously shown that MDM2 binds to and promotes proteasome-mediated degradation of RB and that MDMX binds to and promotes RB degradation in an MDM2-dependent manner [5,6]. Notably, it has been shown that MDM2 binds to and promotes MDMX polyubiquitylation and degradation [27,28]. We thus hypothesized that Beclin 1 might affect the interaction of the MDM2–MDMX–RB complex. As shown in Figure 4e, endogenous Beclin 1–MDM2–MDMX–RB quadruple protein complexes were readily detected in HCT116 or HCT116 p53${}^{-/-}$ cells. Furthermore, ablation of *BECN1* led to a significant increase in the interaction of MDM2 and MDMX (Figure 4f). We next examined the domains involved in the interaction between Beclin 1 and MDM2, MDMX, or RB. As shown in Figure 4g–j, Beclin 1 lacking the BH3 domain significantly reduced its interaction with MDM2 or RB, while it was fully capable of interaction with MDMX. Beclin 1 lacking the BH3 and CCD domains was unable to interact with either MDMX, MDM2, or RB. Furthermore, while Beclin 1 could easily form stable complexes with RB, silencing of MDM2, but not silencing of MDMX, led to significantly reduced Beclin 1 interaction with RB (Figure 4k). Silencing of both MDM2 and MDMX led to even weaker Beclin 1–RB interaction (Figure 4k). Together, these results suggest that the CCD domain of Beclin 1 is involved in MDMX interaction while both the BH3 and CCD domains of Beclin 1 are involved in interaction with MDM2. RB is likely piggy-backed to form the RB–MDM2–MDMX–Beclin 1 complex, in which MDM2–MDMX promotes RB protein degradation, keeping with our previous observation [6]. Ablation of *BECN1* facilitates MDM2–MDMX complex formation and promotes MDMX polyubiquitylation and degradation, consequently leading to RB protein stabilization (Figure 4l).

### 3.5. *Knockdown of BECN1 Suppresses Xenograft Tumor Growth through Activation of RB*

We next investigated the role of the Beclin 1-RB pathway in colorectal cancer growth in xenograft mouse models. HCT116 p53${}^{-/-}$-Antares2 cells [15] bearing *BECN1* shRNA were stably expressed in *RB1* shRNA and were transplanted subcutaneously into flanks of nude mice. As shown in Figure 5a–f, depletion of *BECN1* led to significantly reduced xenograft tumor growth, concomitant with elevated RB and markedly reduced Ki67 (Figure 5d,e). Notably, depletion of *BECN1* also led to reduced autophagy, as evidenced by increased expression of p62 and reduced expression of LC3 aggregates (Figure 5d, insets). Importantly, the knockdown of *RB1* remarkably rescued the tumor growth inhibited by the *BECN1* depletion. Collectively, these results demonstrate that depletion of *BECN1* suppresses colorectal cancer growth, at least in part, via elevation of RB expression.

## 4. Discussion

In the present study, several new findings were observed regarding the role of nuclear Beclin 1 in colorectal cancer. First, the expression of Beclin 1 protein was negatively correlated with tumor suppressor RB protein expression in later stages of colorectal cancers. The second important finding was that nuclear Beclin 1 is critically involved in cell cycle progression through modulation of RB protein stability. Ablation of *BECN1* leads to elevation of RB protein expression, resulting in inhibition of cell cycle progression and retardation of colorectal cancer cell growth in vitro and in vivo. Mechanistically, knockdown of nuclear Beclin 1 promoted MDM2–MDMX interaction, resulting in reduced MDMX expression and upregulated RB expression.

Beclin 1 has an important role in the growth and metastasis of human colorectal cancer. However, the role of Beclin 1 linked to autophagy in CRC development has been extensively studied with conflicting results. Loss of Beclin 1 expression defines poor prognosis by promoting anti-apoptotic pathways, whereas overexpression of Beclin 1, being linked with tumor hypoxia and acidity, also defines subgroups of tumors with aggressive clinical behavior [29]. Meta-analysis indicated that the elevated Beclin 1 expression is associated with tumor metastasis and a poor prognosis in CRC patients [30]. Importantly, in *KRAS*-mutated CRC, nuclear Beclin 1 expression was associated with a significantly decreased OS [31]. Here, we showed that the protein expression of Beclin 1 negatively correlates with OS and RFS of CRC patients, while there is no significant correlation between the *BECN1* mRNA expression and the OS and RFS of CRC patients, which suggests that Beclin 1 protein abundancy is likely influenced by post-translational modifications such as ubiquitylation/deubiquitylation in CRC progression [32]. Since cytoplasmic Beclin 1 is fundamentally important in autophagy, it is conceivable that nuclear Beclin 1 plays an autophagy-independent role in colorectal tumorigenesis. At present, the only known study of nuclear Beclin 1 reported that Beclin 1 directly interacts with DNA topoisomerase IIβ and can be recruited to the DNA double-strand break sites to facilitate DNA repair progression [33]. In this study, we show that nuclear Beclin 1, unable to induce autophagy, can modulate RB protein expression, thereby regulating cell cycle and colorectal cancer cell growth. Beclin 1 functions as an adapter for recruiting multiple proteins in different biological processes. The interaction between Beclin 1 and MDMX has been preliminarily revealed in a cancer-focused protein–protein interactions network research [34], without validation. In this study, we show that Beclin 1 deficiency leads to an elevated RB protein stability in an MDMX-dependent manner. Restoration of nuclear Beclin 1 (Beclin 1${}^{L184A+L187A}$) significantly rescues MDMX expression, which facilitates the degradation of RB.

MDM2 and MDMX are deregulated in many human cancers and exert their oncogenic activity predominantly by inhibiting p53 and RB [35]. MDM2 directly binds to the RBC-pocket, thus preventing RB–E2F interaction and also promoting RB proteasomal degradation [4,5]. MDMX is a close structural analog of MDM2; however, MDMX lacks intrinsic E3 ligase activity. MDM2 binds to MDMX and promotes MDMX proteasomal degradation [27,28]. In our previous study, we showed that MDMX binds to MDM2 and enhances MDM2 ubiquitin E3 ligase activity towards RB, thereby enhancing RB proteasomal degradation [6]. In this work, we showed that ablation of *BECN1* facilitates the interaction between MDM2 and MDMX, resulting in the degradation of MDM2-mediated MDMX degradation.

Autophagy has been reported to exert both tumor-suppressive and tumor-promoting effects in colorectal cancer [9]. Autophagy can prevent the transformation from normal to malignant during the early stages of tumor formation [9]. In this study, we found that elevated Beclin 1 expression correlates with reduced RB protein in advanced stages of colorectal cancer. Silencing of *BECN1* leads to RB-induced G1/S cell cycle arrest, which is independent of autophagy. However, restoration of nuclear Beclin 1 can not fully rescue *BECN1* silencing-mediated reduction of colon cancer cell colony formation. In addition, in the xenograft tumor mouse model, *BECN1* ablation leads to robust inhibition of tumor growth, attributable to both upregulated RB expression and reduced autophagy. Knockdown of *RB1* cannot completely restore tumor growth nor restore autophagy. These results suggest that ablation of *BECN1* inhibits colorectal cancer growth through inhibition of both cell growth and autophagy.

## Figures and Tables

**Figure 1 cancers-14-04735-f001:**
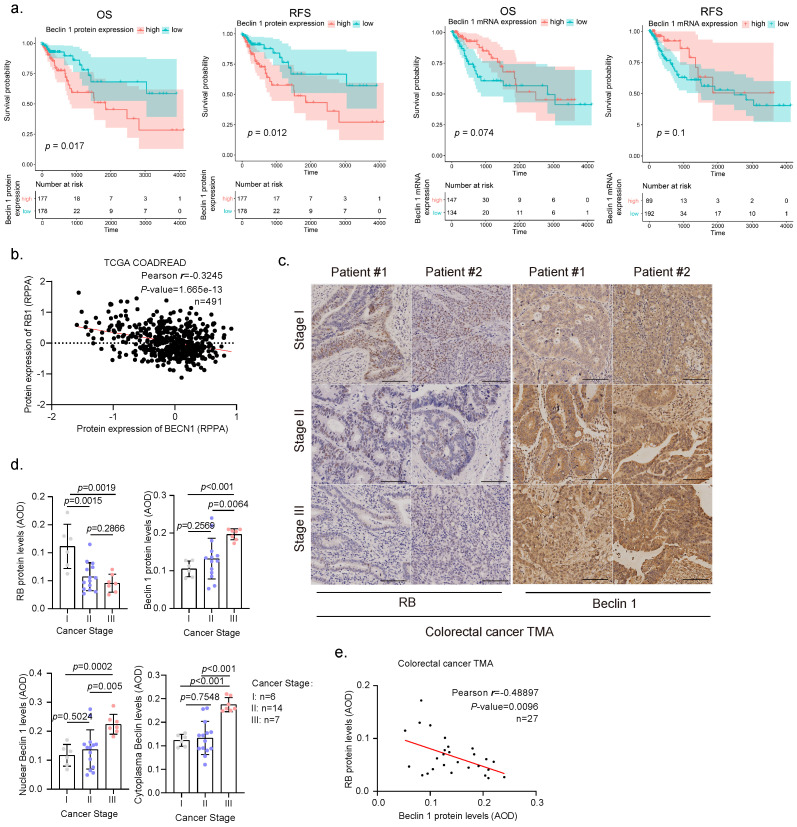
Clinical relevance of Beclin 1 or RB expression in human colorectal cancer specimen. (**a**) Overall survival (OS) and relapse-free survival (RFS) of colon adenocarcinoma patients were based on *BECN1* mRNA expression and protein expression from TCGA-COAD dataset. Patients were stratified into high- or low-expression groups using optimal cutoff from survminer package. The blue and red areas represent the 95% confidence intervals of the high- and low-expressed groups, respectively. The tables listed below the plots show the number of patients at risk at each time point in each group. (**b**) Pearson’s correlation coefficient between the protein expressions of Beclin 1 and RB in the TCGA-COADREAD dataset was obtained from the LinkedOmics database (http://www.linkedomics.org/admin.php (accessed on 10 September 2022)). Reverse-Phase Protein Array, RPPA. (**c**,**d**) Human colorectal cancer tissue microarrays consisting of cancer specimens from three cancer stages (stage I, *n* = 6; II, *n* = 14; III, *n* = 7) were subjected to IHC staining for RB and Beclin 1, with quantitative analyses using average optical density (AOD). (**e**) Pearson’s correlation between RB and Beclin 1 in colorectal cancer patients was analyzed. Scale bar = 50 μm.

**Figure 2 cancers-14-04735-f002:**
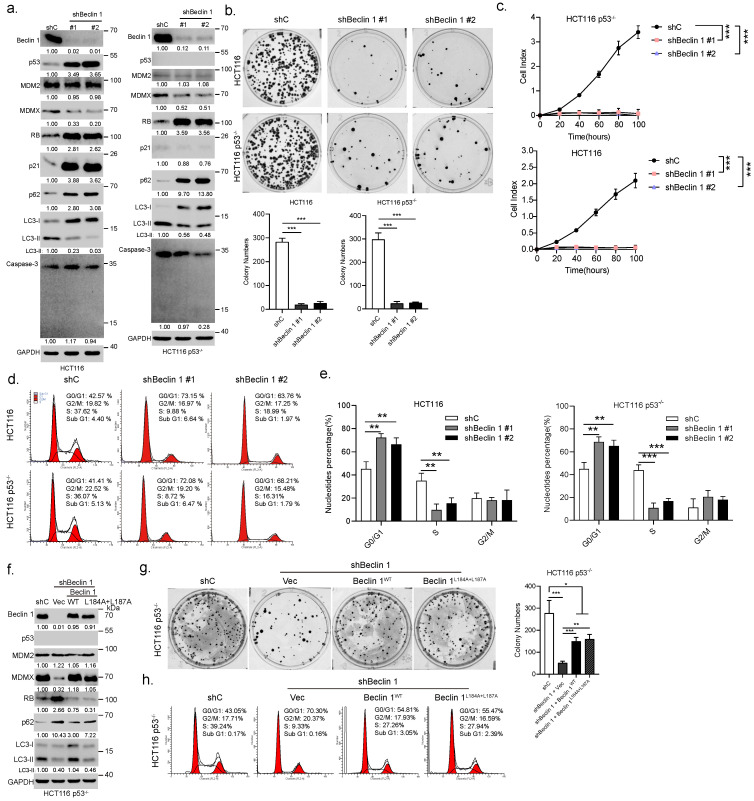
Ablation of *BECN1* leads to cell cycle G1/S arrest and growth inhibition through upregulation of RB protein expression. HCT116 or HCT116 p53${}^{-/-}$ cells stably expressing shRNA specific for *BECN1* (#1 or #2) or control shRNA (shC) were subjected to western blot (**a**), colony formation assays (**b**), real-time cell analyses (RTCA) (**c**), and FACS analyses (**d**). Quantifications of cell cycle derived from FACS analyses (**e**). HCT116 p53${}^{-/-}$ cells stably expressing shBeclin 1 were infected with lentivirus expressing Beclin 1 or Beclin 1${}^{L184A+L187A}$ followed by western blot (**f**), colony formation assays (**g**), and FACS analyses (**h**). Data derived from three independent experiments were presented as mean ± SD. * *p* < 0.05, ** *p* < 0.01, *** *p* < 0.001. The uncropped blots and molecular weight markers are shown in Appendix A.

**Figure 3 cancers-14-04735-f003:**
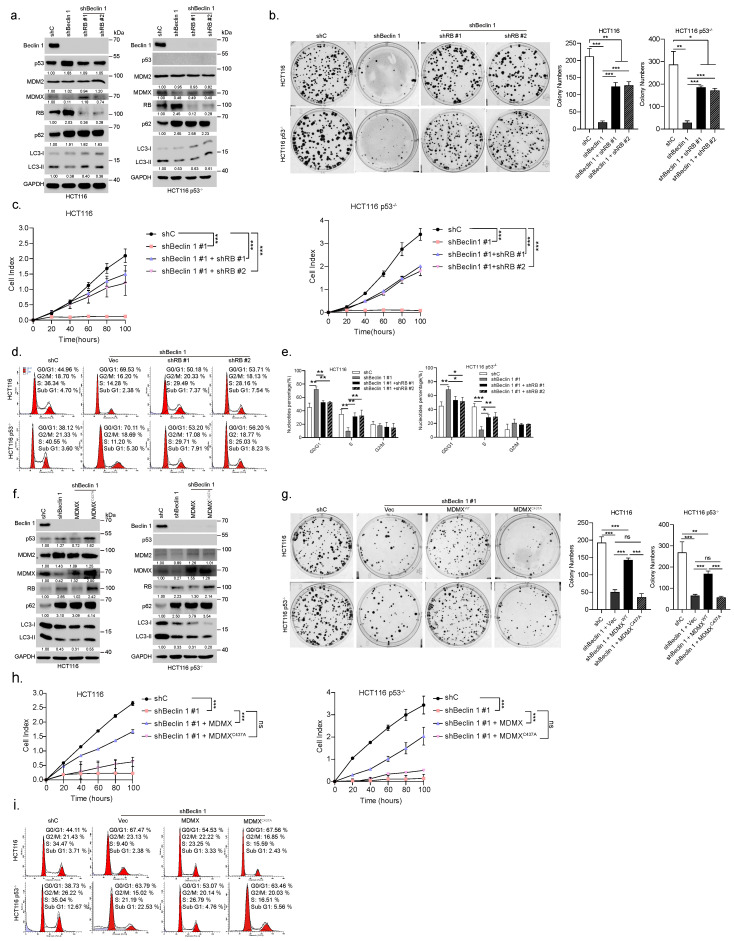
Ablation of *BECN1* reduces MDMX expression resulting in the RB-dependent cell cycle G1/S arrest and growth inhibition. HCT116 or HCT116 p53${}^{-/-}$ cells stably expressing shBeclin 1 were infected with lentivirus shRNA specific for *RB1* (#1 or #2) or control shRNA and subjected to western blot (**a**), colony formation assays (**b**), real-time cell analyses (RTCA) (**c**), and FACS analyses (**d**). Quantifications of cell cycle derived from FACS analyses (**e**). HCT116 or HCT116 p53${}^{-/-}$ cells stably expressing shBeclin 1 were infected with lentivirus expressing MDMX or MDMX${}^{C437A}$ and subjected to western blot (**f**), colony formation assays (**g**), real-time cell analyses (RTCA) (**h**), and FACS analyses (**i**). Data were derived from three independent experiments and were presented as mean ± SD. * *p* < 0.05, ** *p* < 0.01, *** *p* < 0.001. The uncropped blots and molecular weight markers are shown in Appendix A.

**Figure 4 cancers-14-04735-f004:**
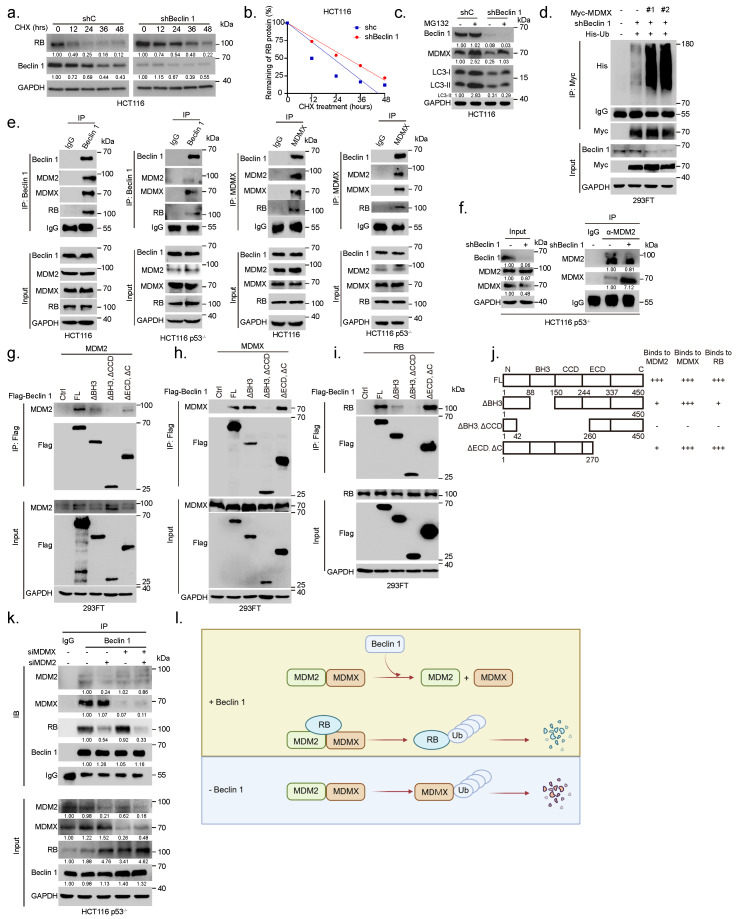
Ablation of *BECN1* facilities MDM2–MDMX interaction to promote MDMX degradation and RB protein stabilization. (**a**) HCT116 cells stably expressing shBeclin 1 or control shRNA (shC) were treated with 50 μg/mL cycloheximide (CHX) for the indicated time intervals. (**b**) The plots of RB protein half-life. (**c**) HCT116 cells stably expressing shBeclin 1 or control shRNA (shC) were treated with 20 μM MG132 for 12 h before collection. Total cell lysates were subjected to western blot. (**d**) 293FT cells stably expressing shC or shBeclin 1 (#1 or #2) were transfected with expressing plasmids of Myc-MDMX, His-Ub. Then, cells were treated with 20 μM MG132 for 6 h before collection. Ubiquitylation of MDMX was examined by IP-western analyses. (**e**) HCT116 or HCT116 p53${}^{-/-}$ cells were subjected to co-immunoprecipitation (co-IP) using either a specific antibody for Beclin 1 and MDMX or IgG control followed by western blot analyses. (**f**) HCT116 p53${}^{-/-}$ cells stably expressing shC or shBeclin 1 were subjected to co-immunoprecipitation (co-IP) assay using a specific antibody for MDM2 or IgG control followed by western blot analyses as indicated. (**g**–**j**) 293FT cells were co-transfected with Flag-Beclin 1 (FL) or an indicated Beclin 1 deletion construct and MDM2 (**g**), MDMX (**h**), or RB (**i**). Total cell lysates were subjected to IP-western analyses. (**j**) A schematic illustration shows MDM2–Beclin 1, MDMX–Beclin 1, and RB–Beclin 1 interaction domains. (**k**) HCT116 p53${}^{-/-}$ cells transfected with siMDM2, siMDMX, siMDM2/siMDMX were subjected to co-immunoprecipitation (co-IP) using a specific antibody followed by western blot analyses as indicated. (**l**) A sketch depicts a model of the influence of Beclin 1 on quadruple protein complexes formation among Beclin 1, RB, MDMX, and MDM2. The uncropped blots and molecular weight markers are shown in Appendix A.

**Figure 5 cancers-14-04735-f005:**
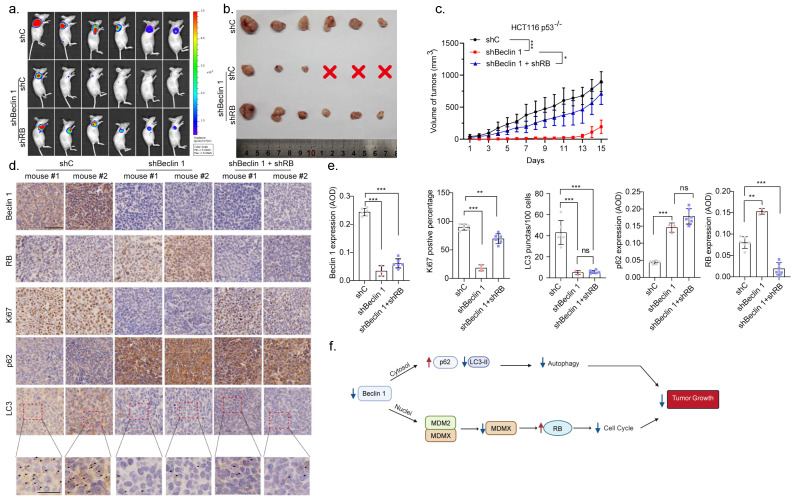
Knockdown of *BECN1* suppresses xenograft tumor growth, which is reversed by simultaneous ablation of *RB1*. (**a**) For tumor growth assay, HCT116 p53${}^{-/-}$-Antares2 cells (5 × 10${}^{6}$) stably expressing shBeclin 1 or shBeclin 1+shRB were subcutaneously transplanted into nude mice (*n* = 6/group). Diphenylterazine (DTZ) was injected into tumor regions in anesthetized mice and bioluminescence was subsequently imaged using Caliper IVIS Lumina III. (**b**) The harvested tumors were photographed. (**c**) The tumor volumes of cell-derived xenografts were determined every other day. (**d**) The tumor sections were subjected to immunohistochemistry (IHC) staining for Beclin 1, RB, Ki67, p62, and LC3 (Scale bar = 50 μm; Inset scale bar = 10 μm). (**e**) The respective quantification data were presented as mean ± SEM. * *p* < 0.05, ** *p* < 0.01, *** *p* < 0.001. (**f**) A model depicts that nuclear Beclin 1 interferes with MDM2–MDMX interaction to stabilize MDMX, leading to the RB protein instability, cell cycle progression, and ultimately tumor growth.

## Data Availability

The data that support the findings of this study are available from the corresponding author upon reasonable request.

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
