# Peer review of "Nuclear Beclin 1 Destabilizes Retinoblastoma Protein to Promote Cell Cycle Progression and Colorectal Cancer Growth"

_cancers, 2022, doi:10.3390/cancers14194735_

Round 1
Reviewer 1 Report
Figure 1b, I also check expression correlation between RB(RB1) and Beclin-1(BECN1) in TCGA dataset in Colon cancer. I used cbioportal and gepia website. I found RB and beclin-1 are positive correlation in colon cancer, which is completely opposite to authors' conclusion. What is more, according to authors' conclusion, beclin-1 expression should be negative correlated with colon cancer patient survival, but I check TCGA dataset, no change.
In this case, in this research, the hypothesis that author base on is completely wrong.
Author Response
Response: This is important issue regard to the correlation of expression between RB (RB1) and Beclin 1 (BECN1) in colon cancer and the clinical relevance. To address the reviewer’s concerns, we re-performed expression correlation analyses using the updated database of cBioPortal, GEPIA, and GDC TCGA COAD. The results reveal that the mRNA levels of RB1 and BECN1 have a weak positive correlation (r<0.3) (Turk J Emerg Med, 2018, PMID: 30191186) (revised supplementary Figure S1b-d), in keeping with the analyses from the reviewer. We then further analyzed the correlation between Beclin 1 and RB protein expression of colorectal cancer patients in TCGA COADREAD dataset obtained from the LinkedOmics (Nucleic Acids Res, 2018, PMID: 29136207). In sharp contrast to mRNA levels that show a weak positive correlation, our analyses reveal a moderate negative correlation (r= -0.3245, Pearson’s correlation) between RB and Beclin 1 proteins (revised Figure 1b). Together with our analyses in this study using the human colon cancer specimens (tissue microarrays, TMA), which reveal a negative correlation between RB and Beclin 1 protein expression (Figure 1c-1e), these results strongly suggest that protein expressions of RB and Beclin 1 are negatively correlated in CRC.
In addition to the correlation between RB expression and CRC patients overall survival (OS) (original Figure 1a), we also analyzed the overall survival (OS) and relapse-free survival (RFS) of human colon adenocarcinoma patients based on Beclin 1 protein or mRNA expression using the TCGA-COAD dataset. These analyses reveal that the protein expression of Beclin 1 negatively correlated with OS and RFS of CRC patients, while there was no correlation between the BECN1 mRNA expression and OS and RFS of CRC patients (revised Figure 1a). We have integrated these results in the sections of Results and Discussion (revised lines 297-301).
Reviewer 2 Report
This manuscript by Pan and colleagues highlights the role of nuclear beclin1 in controlling tumor growth in CRC model. The findings are well defined with the experimental control and of interest in relation to understanding the mechanism of inhibition of colorectal cancer growth via RB mediated cell growth and autophagy. The manuscript is well written and can be considered for publication if the following comments are addressed.
1. 1. CRC is a heterogeneous disease, with subtypes that are driven by different types of genomic alterations. In retrospect of this, did authors verified validated their finding other CRC cells line like; CL40, SW1417, and others that are considered suitable as models for CRC tumors? Can author’s highlight and address this point in their study section.
2. 2. Role of Beclin is clearly established in regulating apoptosis. Especially, when autophagy-inducing activity of Beclin 1 is inhibited by the overexpression of Bcl-2 family members. Did authors look at the levels of Bcl/bax/MCL members or PI3K?
3. 3. Why did the authors choose to do flank injection for in-vivo studies? Orthotopic injection would have been the closest. Further, liver metastasis is common in CRC patients. Given that, did authors observed any difference in metastasis of liver with WT and Beclin knockdown mouse?
Author Response
Reviewer 2 comment #1:
CRC is a heterogeneous disease, with subtypes that are driven by different types of genomic alterations. In retrospect of this, did authors verified validated their finding other CRC cells line like; CL40, SW1417, and others that are considered suitable as models for CRC tumors? Can author’s highlight and address this point in their study section.
Response: We appreciate the reviewer’s suggestion to verify the findings using other CRC cells. Due to the short revision time, we took advantage of the CRC Caco-2, routinely maintained in this laboratory. As shown in the revised supplementary Figure S1e, ablation of BECN1 significantly increased RB expression in Caco-2 cells. Interestingly, ablation of Beclin 1 could also lead to upregulation of RB expression in human lung adenocarcinoma A549 cells.
Reviewer 2 comment #2:
Role of Beclin is clearly established in regulating apoptosis. Especially, when autophagy-inducing activity of Beclin 1 is inhibited by the overexpression of Bcl-2 family members. Did authors look at the levels of Bcl/bax/MCL members or PI3K?
Response: We appreciate the reviewer’s suggestion. To address whether knockdown of Beclin 1-mediated reduction of cancer cell growth is attributed to apoptosis in our experimental system, we performed additional western blotting analyses. Our results showed that knockdown of BECN1 did not induce apoptosis, as evidenced by undetectable cleaved caspase 3 (revised Figure 2a). In addition, silencing of BECN1 also did not significantly increase the subG1 population shown by FACS (revised Figure 2d, 2h, 3d, 3i). These data indicate that silencing of BECN1 does not promote apoptosis in HCT116 cells.
Reviewer 2 comment #3:
Why did the authors choose to do flank injection for in-vivo studies? Orthotopic injection would have been the closest. Further, liver metastasis is common in CRC patients. Given that, did authors observed any difference in metastasis of liver with WT and Beclin knockdown mouse?
Response: We appreciate the reviewer’s insightful comments. We are aware that orthotopic injection is widely used for CRC metastasis research (Nat Commun, 2022, PMID: 35987910; Nat Commun, 2022, PMID: 36008411). In this study, we showed that ablation of Beclin 1 leads to elevation of RB protein expression, resulting in the inhibition of cell cycle progression and retardation of colorectal cancer cell growth in vitro and in vivo using a conventional xenograft tumor growth mouse model. We have not addressed colon cancer liver metastasis in this study, which deserves further investigation.
Reviewer 3 Report
In this manuscript, Pan et al., study entitled on "Nuclear Beclin1 destabilizes retinoblastoma protein to promote cell cycle progression and colorectal cancer growth". Author concluded the results that expression of nuclear Beclin1 is upregulated in colorectal cancer and Beclin1 silenced cells are upregulated RB protein expression resulting in cell cycle arrest and colorectal cancer cells growth inhibition. Beclin 1 ablation was inhibited in xenograft tumor growth that was reversed by simultaneous silencing of RB. Moreover, Author observed that knockdown of Beclin 1 promoted MDM2 and MDMx interaction resulting in instability of protein MDMx and stability of RB protein. However i have some minor points that must be addressed before paper will be suitable for publication.
In results,
fig.2g colony formation assay need to be quanitified.
As you mentioned in the introduction that beclin1 is localized in both cytoplasm and nucleus. fig.1c tissue micro array quantification results showed total and nuclear beclin 1 optical density. what is the cytoplasic level of beclin 1 expression and what is the role of cytosolic beclin1 in this study.
Cytoplasmic Beclin 1 functions as a core protein in class III PI3K complexes that mediate different membrane trafficking events in autophagy. is there any interaction that nuclear beclin 1 has with cytoplasic beclin1.
Ablation of BECN1 increases RB protein expression accompanied by cell cycle G1/S arrest and growth inhibition. have you done apoptosis analysis.
The manuscript is well written but could use some proofreading and method section is aptly described with necessary details. The manuscript should be accepted after minor revision.
ps. supplimentary file unable to view due to different format.
Author Response
Reviewer 3 comment #1:
In results, fig.2g colony formation assay need to be quanitified.
Response: Thanks for the suggestion. Accordingly, we have quantified the data derived from colony formation assays (revised Figure 2b, 2g,3b,3g).
Reviewer 3 comment #2:
As you mentioned in the introduction that beclin1 is localized in both cytoplasm and nucleus. fig.1c tissue micro array quantification results showed total and nuclear beclin 1 optical density. what is the cytoplasic level of beclin 1 expression and what is the role of cytosolic beclin1 in this study.
Response: Thanks for the suggestion. Accordingly, we quantified the cytoplasmic Beclin 1 protein levels, which was also significantly increased in advanced CR specimens (revised Figure 1d). With regard to the role of cytoplasmic Beclin 1, we think that the autophagic-promoting function of cytosolic Beclin1 plays an important role in colon cancer cell growth and tumor growth. In this study, restoration of nuclear Beclin 1 is unable to fully rescue Beclin 1 silencing-mediated reduction of colon cancer cell coloniality (revised Figure 2g). In addition, in the xenograft tumor mouse model, Beclin 1 ablation leads to robust inhibition of tumor growth, attributable to both up-regulated RB expression and reduced autophagy. Knockdown of RB is unable to completely restore tumor growth nor restore autophagy. These results strongly suggest that cytoplasmic Beclin 1 plays an important role in promoting autophagy to facilitate colorectal cancer growth.
Reviewer 3 comment #2:
Cytoplasmic Beclin 1 functions as a core protein in class III PI3K complexes that mediate different membrane trafficking events in autophagy. is there any interaction that nuclear beclin 1 has with cytoplasic beclin1.
Response: This is an interesting point. Considering the different subcellular localization, it is conceivable that nuclear Beclin 1 might be able to interact with cytoplasmic proteins during nuclear breakdown in mitosis.
Reviewer 3 comment #3:
Ablation of BECN1 increases RB protein expression accompanied by cell cycle G1/S arrest and growth inhibition. have you done apoptosis analysis.
Response: We appreciate the reviewer’s suggestion. During revision, we performed additional western blotting analyses. Our results showed that knockdown of Beclin 1 did not induce apoptosis, as evidenced by undetectable cleaved caspase 3 (revised Figure 2a). In addition, silencing of BECN1 also did not significantly increase the subG1 population shown by FACS (revised Figure 2d, 2h, 3d, 3h). These data indicate that silencing of BECN1 does not promote apoptosis in HCT116 cells.
Reviewer 3 comment #4:
The manuscript is well written but could use some proofreading and method section is aptly described with necessary details. The manuscript should be accepted after minor revision.
- supplimentary file unable to view due to different format.
Response: We appreciate the reviewer’s suggestion. Accordingly, we added details about colony formation quantification, correlation and survival analyses, and statistical analysis in the Method section in the revised manuscript (revised lines 122-123; lines 150-158; lines 162-163). The supplementary file in the same format was uploaded during the submission of the revised manuscript.
Round 2
Reviewer 1 Report
As authors have remodeled their paper and addressed my questions properly, I do not have further suggestions.